# *MYO5A* Frameshift Variant in a Miniature Dachshund with Coat Color Dilution and Neurological Defects Resembling Human Griscelli Syndrome Type 1

**DOI:** 10.3390/genes12101479

**Published:** 2021-09-23

**Authors:** Matthias Christen, Madeleine de le Roi, Vidhya Jagannathan, Kathrin Becker, Tosso Leeb

**Affiliations:** 1Institute of Genetics, Vetsuisse Faculty, University of Bern, 3001 Bern, Switzerland; matthias.christen@vetsuisse.unibe.ch (M.C.); vidhya.jagannathan@vetsuisse.unibe.ch (V.J.); 2Department of Pathology, University of Veterinary Medicine Hannover, 30559 Hannover, Germany; madeleine.de.le.roi@tiho-hannover.de (M.d.l.R.); kathrin.a.becker@basf.com (K.B.)

**Keywords:** *Canis lupus familiaris*, animal model, neurology, dermatology, precision medicine

## Abstract

A 1-month-old, female, smooth-haired miniature Dachshund with dilute color and neurological defects was investigated. The aim of this study was to characterize the clinical signs, histopathological changes and underlying genetic defect. The puppy had visible coat color dilution and was unable to hold its head on its own or to remain in a stable prone position for an extended period. Histopathological examination revealed an accumulation of clumped melanin and deposition of accumulated keratin within the hair follicles, accompanied by dermal pigmentary incontinence. These dermatological changes were compatible with the histopathology described in dogs with an *MLPH*-related dilute coat color. We sequenced the genome of the affected dog and compared the data to 795 control genomes. *MYO5A*, coding for myosin VA, was investigated as the top functional candidate gene. This search revealed a private homozygous frameshift variant in *MYO5A*, XM_022412522.1:c.4973_4974insA, predicted to truncate 269 amino acids (13.8%) of the wild type myosin VA protein, XP_022268230.1:p.(Asn1658Lysfs*28). The genotypes of the index family showed the expected co-segregation with the phenotype and the mutant allele was absent from 142 additionally genotyped, unrelated Dachshund dogs. *MYO5A* loss of function variants cause Griscelli type 1 syndrome in humans, lavender foal in horses and the phenotype of the *dilute* mouse mutant. Based on the available data, together with current knowledge on other species, we propose the identified *MYO5A* frameshift insertion as a candidate causative variant for the observed dermatological and neurological signs in the investigated dog.

## 1. Introduction

Griscelli syndrome (GS, OMIM #214450) represents a group of rare diseases in humans with a monogenic autosomal recessive mode of inheritance. Griscelli et al. reported the first patients in 1978 [1]. The authors described two patients with partial albinism, frequent pyogenic infections and acute episodes of fever, neutropenia and thrombocytopenia. Three distinct types of GS, with the common feature of silvery grey hair, pigmentary dilution of the skin and melanin clumps within hair shafts, have been described in human medicine. In GS type 1, variants in the *MYO5A* gene, coding for myosin VA, cause severe primary neurologic impairment in addition to the aforementioned dermatological changes [2,3,4] (OMIM #214450). GS type 2 is caused by variants in *RAB27A* and characterized by pigment dilution in combination with immunological abnormalities [5] (OMIM #607624). GS type 3 is due to variants in *MLPH* and comprises an isolated pigment dilution without any defects in the nervous or immune systems [6] (OMIM #609227).

The pigment dilution in GS is the result of impaired melanosome transport within melanocytes [7]. Perinuclear melanosomes are transported along actin filaments to the periphery and tethered via myosin VA to the dendritic tips of the cell [8]. In the absence of wild type myosin VA in GS type I patients, this binding step cannot occur and melanosomes are not correctly recruited to the cell periphery. As a result, the transfer of melanosomes from melanocytes to keratinocytes and into growing hair shafts is hindered [9].

Furthermore, myosin VA-mediated transport is also required in neurons and especially in the cerebellum. There, the motor protein myosin VA pulls the endoplasmic reticulum into the dendritic spines of Purkinje neurons [10]. The endoplasmic reticulum is important for Ca^2+^ storage and essential for intracellular Ca^2+^ signaling [11,12]. Consequently, missing myosin VA-mediated transport reduces synaptic plasticity and leads to a neurological phenotype [10] (OMIA 001501-9796).

Analogous to the human phenotype, *Myo5a* loss-of-function variants are responsible for the *dilute* mouse mutant [13] and the *dilute-lethal* rat mutant [14]. Moreover, a single base pair deletion in exon 30 of the *MYO5A* gene causes lavender foal syndrome in horses, which is phenotypically similar to human GS type 1 [15].

This study was initiated after a smooth-haired miniature Dachshund with a striking coat color dilution and neurological deficits was reported. The goal of the study was to characterize the phenotype of the puppy and to investigate a possible underlying causative genetic defect.

## 2. Materials and Methods

### 2.1. Clinical Examination

One affected 1-month-old smooth-haired miniature Dachshund puppy was investigated. Clinical examination was carried out by a veterinary clinician. EDTA blood samples from the patient, five unaffected full siblings and their parents were collected for genomic DNA isolation.

### 2.2. Necropsy, Histopathology, and Immunohistochemistry

A full necropsy of the affected puppy was performed and routinely collected organ samples were fixed in 10% neutral-buffered formalin. Tissue samples were trimmed and embedded in paraffin. For histological examination, organ samples were cut into 4–5 µm thick sections and subsequently stained with hematoxylin and eosin (HE). Immunohistochemistry was performed to visualize axonal damage by using an antibody directed against amyloid precursor protein. After deparaffinization, blocking of endogenous peroxidase and antigen retrieval, signal detection was subsequently achieved to incubation with a primary and a secondary antibody by using the avidin-biotin-peroxidase complex and 3′-diaminobenzidintetrahydrochlorid. Afterwards, sections were counterstained with Mayer’s hemalum.

### 2.3. Control Samples for Genetic Analyses

In addition to the investigated family, 142 blood samples from Dachshunds, which had been donated to the Vetsuisse Biobank, were used. They represented unrelated population controls without reports of a similar phenotype.

### 2.4. DNA Extraction

Genomic DNA was isolated from the EDTA blood with the Maxwell RSC Whole Blood Kit using a Maxwell RSC instrument (Promega, Dübendorf, Switzerland).

### 2.5. Whole-Genome Sequencing

An Illumina TruSeq PCR-free DNA library with ~500 bp insert size of the affected dog was prepared. We collected 296 million 2 × 150 bp paired-end reads on a NovaSeq 6000 instrument (32.8 × overage). Mapping to the CanFam3.1 reference genome assembly was performed as described [16]. A graphical overview of the bioinformatics pipeline is shown in Appendix A. The sequence data were deposited under study accession PRJEB16012 and sample accession SAMEA8157168 at the European Nucleotide Archive.

### 2.6. Variant Calling

Variant calling was performed using GATK HaplotypeCaller [17] in gVCF mode as described [16]. For private variant filtering, we used control genome sequences from 786 dogs from genetically diverse breeds and 9 wolves. These genomes either were publicly available [18] or produced during other previous projects (Appendix A). To predict the functional effects of the called variants, SnpEff [19] software, together with the CanFam3.1 reference genome assembly and NCBI Annotation Release 105, was used (Appendix A).

### 2.7. Gene Analysis

Numbering within the canine *MYO5A* gene corresponds to the NCBI RefSeq accession numbers XM_022412522.1 (mRNA) and XP_022268230.1 (protein).

### 2.8. PCR and Sanger Sequencing

The *MYO5A*:c.4973_4974insA variant was genotyped by direct Sanger sequencing of PCR amplicons. The PCR product was amplified from genomic DNA using AmpliTaqGold360Mastermix (Thermo Fisher Scientific, Waltham, MA, USA), together with primers 5′-AGA GAA GTG GGC CTT CTG GT-3′ (Primer F) and 5′-GAG CTT CCA AGC CAC TTC TG-3′ (Primer R). After treatment with exonuclease I and alkaline phosphatase, PCR amplicons were sequenced on an ABI 3730 DNA Analyzer (Thermo Fisher Scientific, Waltham, MA, USA). Sanger sequences were analyzed using the Sequencher 5.1 software (Gene Codes, Ann Arbor, MI, USA).

## 3. Results

### 3.1. Clinical Examination

In a litter of smooth-haired miniature Dachshunds with two male and four female offspring, one of the female puppies had a striking dilute coat color. This female puppy was presented at four weeks of age to the veterinarian; it had to be kept separate from its siblings for a few days prior to presentation because its siblings had started to nibble at it. The mother, however, was still taking care of the puppy as usual. The owner reported that the puppy did not show any urge to seek the proximity of its siblings and did not whimper when separated. When it was put in the run with its siblings, it fell directly on its side and then rowed with the upper front leg. It managed to roll over to the other side but did not manage to hold a normal prone position. It hardly reacted to environmental stimuli.

On examination, the puppy weighed 0.82 kg and its external features were normally developed; no abnormalities compared to its siblings were found, except for the visible coat color dilution (Figure 1). Upon handling, the puppy could not maintain an upright head position compared to her littermates. The head had to be held at all times, even during feeding. Palpatory tension in the neck and shoulder area was present, but not enough to support the head or coordinate its movement. The puppy was euthanized due to the severity of the clinical phenotype.

### 3.2. Gross, Histopathological, and Immunohistochemical Findings

Grossly, the fur of the puppy was diffusely light red in color. Other macroscopical changes included a V-shaped bend, 45 degrees dorsocaudally, of the distal portion of the spleen and minor agonal changes in different organs. Histopathological examination revealed multifocal accumulation of melanin and deposition of clumped keratin in the follicular epithelium of haired skin (Figure 2). Furthermore, a mild, multifocal, dermal pigmentary incontinence was observed (Figure 2). Immunohistochemical investigation for the presence of amyloid precursor protein as a marker for axonal damage did not reveal alterations within the central nervous system.

### 3.3. Genetic Analysis

We sequenced the genome of the affected dog and searched for homozygous variants in the candidate gene *MYO5A* that were not present in the genome sequences of the 786 control dogs and nine wolves (Table 1 and Appendix A).

This analysis identified a single homozygous private protein-changing variant in the investigated candidate gene. The variant, an insertion of an adenine within the coding sequence, can be designated as Chr30:18,004,551_18;004,552insT (CanFam3.1 assembly). It is a frameshift variant, XM_022412522.1:c.4973_4974insA, predicted to truncate 269 codons encoding the C-terminus of the wild type myosin VA protein, XP_022268230.1:p.(Asn1658Lysfs*28). We did not investigate whether any mutant protein is expressed or whether the premature stop codon caused by the frameshift variant leads to nonsense-mediated decay of the transcript. We confirmed the presence of the frameshift variant in *MYO5A* by Sanger sequencing and genotyped the index family, as well as the 142 control Dachshund dogs (Figure 3).

The case was homozygous for the mutant allele, while none of the 142 unrelated control dogs carried this allele. The genotypes at the insertion co-segregated with the investigated phenotype in the family as expected for a monogenic autosomal recessive mode of inheritance (Figure 3b). The parents, as well as two healthy siblings of the affected dog, carried the variant allele in a heterozygous state.

## 4. Discussion

In dogs, coat color dilution that affects the distribution of both eumelanin and pheomelanin has so far only been explained by variants in the *MLPH* gene [20,21,22]. Initially, the presented case herein caught the breeders’ attention only because of the dilution of the coat color. Neurological signs subsequently became apparent. The puppy resembled two previously described cases of Rhodesian Ridgebacks that also showed pigment dilution in combination with neurological signs [23]. However, the genetic basis for the phenotype in the Rhodesian Ridgebacks was not investigated.

The clinical phenotype of the described smooth-haired miniature Dachshund showed a striking resemblance to previously-reported cases of GS type 1 in humans and lavender foal syndrome in horses [4,15].

Histopathological alterations of the skin matched what has been described in dogs with *MLPH*-related dilute coat color [24]. Loss of function of *MLPH* leads to isolated coat color dilution without neurological signs. However, *MLPH* mutant dogs are predisposed to developing color dilution alopecia [24,25]. Alopecia was not observed in the present case; however, color dilution alopecia typically manifests between 4 months and 3 years of age.

The neurological impairment was not associated with morphologic changes at the gross or histopathological level. However, molecular changes such as impairment of Ca^2+^ storage are not necessarily accompanied by morphologic alterations.

The misshaped spleen probably represents an autonomous congenital malformation with minor clinical relevance. However, a correlation with the observed *MYO5A* variant cannot be excluded with certainty.

The *MYO5A* gene encodes for myosin VA, an intracellular organelle transport protein with important functions in the dendritic spines of melanocytes and Purkinje cells [7,10]. Our analysis revealed a private homozygous frameshift variant, *MYO5A*:c.4973_4974insA, in the studied Dachshund. The frameshift led to a premature termination codon and was predicted to truncate the coding sequence for the C-terminus of the wild type myosin VA protein, including parts of the highly conserved globular tail domain. This domain represents the cargo-binding domain of myosin VA [26]. Therefore, the identified *MYO5A* frameshift variant may be assumed to cause a complete loss-of-function allele. Together with the knowledge of the effects of *MYO5A* variants in other species, these data suggest *MYO5A*:c.4973_4974insA as a candidate causative genetic variant for the phenotype in the investigated puppy.

The phenotype of the affected dog closely resembles the phenotype of human patients with Griscelli syndrome type I. The identification of a candidate causative variant enables genetic testing and the detection of heterozygous carriers so that further unintentional breeding of affected dogs can be prevented. If similarly affected Dachshunds should appear, the genetic test can also be used to quickly confirm the suspected diagnosis.

## Figures and Tables

**Figure 1 genes-12-01479-f001:**
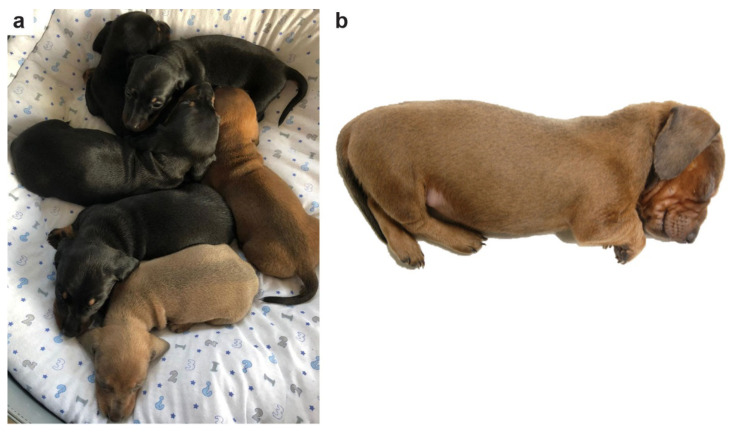
Coat color dilution phenotype. (**a**) The photo shows the entire litter consisting of six puppies. Four of the non-affected puppies had black and tan coat colors. The affected puppy at the bottom of the photo had a dilute red color, which is much lighter than the standard red color of its non-affected sibling on the right. (**b**) Larger photo from the affected puppy alone.

**Figure 2 genes-12-01479-f002:**
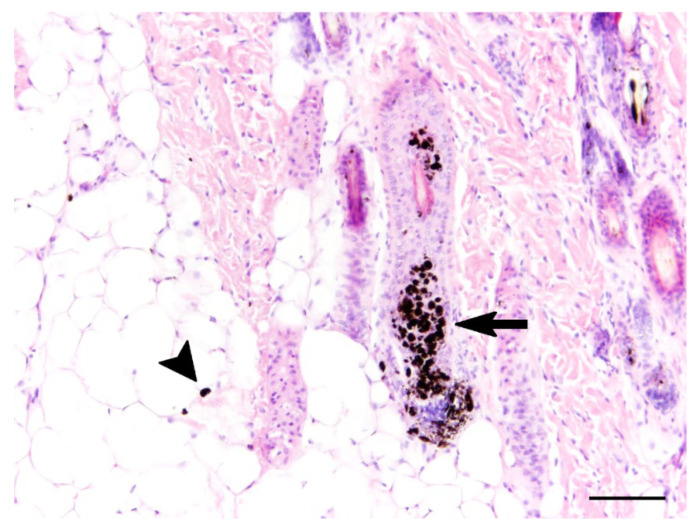
Histopathology of the skin. Multifocally, epithelial cells of hair follicles displayed an accumulation of clumped melanin (arrow) and deposition of accumulated keratin. In the dermis, a mild, multifocal pigmentary incontinence (arrowhead) was present. Hematoxylin and eosin stain (bar = 100 µm).

**Figure 3 genes-12-01479-f003:**
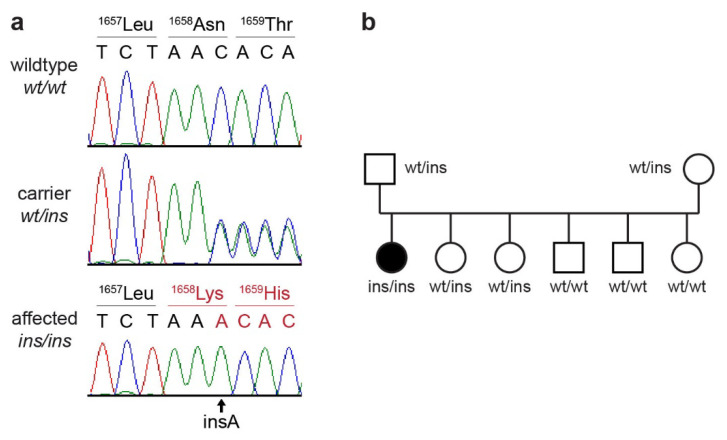
Details of the *MYO5A*:c.4973_4974insA variant. (**a**) Representative Sanger sequencing chromatograms of dogs with the three different genotypes. A homozygous insertion of a single adenine is visible in the affected dog. (**b**) The genotypes in the Dachshund family showed the expected co-segregation with the phenotype in the index family.

**Table 1 genes-12-01479-t001:** Results of variant filtering in the affected miniature Dachshund dog against 795 control genomes.

Filtering Step	Homozygous Variants
All variants in the affected miniature Dachshund	2,698,983
Private variants	1688
Protein-changing ^1^ private variants	12
Protein-changing ^1^ private variants in *MYO5A*	1

^1^ “Protein-changing” variants have a SnpEff predicted moderate or high impact [19]. These include missense, nonsense, frameshift, and splice site variants among others.

## Data Availability

The accessions for the sequence data reported in this study are listed in Appendix A.

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
