# Peer review of "MYO5A Frameshift Variant in a Miniature Dachshund with Coat Color Dilution and Neurological Defects Resembling Human Griscelli Syndrome Type 1"

_genes, 2021, doi:10.3390/genes12101479_

Round 1

Reviewer 1 Report

Very impressive, well written paper. Perfect English and very concise - a pleasure to read. Particularly appreciative of the 2.7 gene Analysis section as many people fail to identify the refseq accession number.

My only suggestions are:
1. To define the specific type of dachshund (miniature, long/short haired, smooth/wire haired etc.). Although, I'm not sure how exactly Dachshund subtypes are categorised within the Swiss/German kennel clubs and therefore may not be suitable.
2. In the conclusion/discussion - identify how exactly this paper adds to the field of veterinary medicine/genomic medicine. e.g. Allowing affected dogs to be diagnosed. Or used as part of a genomic test in breeding programmes to avoid this disease in subsequent generations

Author Response

(1)

Very impressive, well written paper. Perfect English and very concise - a pleasure to read. Particularly appreciative of the 2.7 gene Analysis section as many people fail to identify the refseq accession number.

Response: Thank you very much for the kind words and compliments.

(2)

My only suggestions are: To define the specific type of dachshund (miniature, long/short haired, smooth/wire haired etc.). Although, I'm not sure how exactly Dachshund subtypes are categorised within the Swiss/German kennel clubs and therefore may not be suitable.

Response: According to FCI standard, these were smooth-haired miniature Dachshunds. We added this information.

(3)

In the conclusion/discussion - identify how exactly this paper adds to the field of veterinary medicine/genomic medicine. e.g. Allowing affected dogs to be diagnosed. Or used as part of a genomic test in breeding programmes to avoid this disease in subsequent generations.

Response: We expanded the last sentence (conclusion) of the manuscript accordingly.

Reviewer 2 Report

This is a very interesting and well-presented manuscript. It concerns a rare syndrome in canines which resembles some well-documented human and other animal syndromes. The genomic and genetic analyses are clear and well designed and the results are well presented.

However, I have some minor comments/questions for the authors:

In the introduction section, there are a lot of information about Griscelli syndrome in humans particularly, some references about mouse and rats and almost none information about canines. These details about canines are presented in results and discussion (e.g. lines 115-122, 177-186 etc) and they should be moved accordingly, so the reader can understand from the introduction what is the subject of the MS. 

2.5 and 2.5 describe the bioinformatics analysis and you cite [16] in both; I think you should provide some details about the pipeline (maybe in a graphical way) so the reader can follow

Lines 101-103 can be removed

Table 2: Protein-changing variants is not a useful term. Probably "Variants altering the protein sequence"  or something similar

Did you check for any other mutations in the genes associated with other types of GS?

Author Response

(1)

This is a very interesting and well-presented manuscript. It concerns a rare syndrome in canines which resembles some well-documented human and other animal syndromes. The genomic and genetic analyses are clear and well designed and the results are well presented.

Response: Thank you very much for the kind words and compliments.

(2)

In the introduction section, there are a lot of information about Griscelli syndrome in humans particularly, some references about mouse and rats and almost none information about canines. These details about canines are presented in results and discussion (e.g. lines 115-122, 177-186 etc) and they should be moved accordingly, so the reader can understand from the introduction what is the subject of the MS.

Response: To the best of our knowledge, no report of a dog with confirmed MYO5A variant existed in the scientific literature prior to our manuscript. We therefore characterized the phenotype of such a dog as part of this manuscript. The goal of the study is clearly stated at the end of the introduction (“The goal of the study was to characterize the phenotype of the puppy and to investigate a possible underlying causative genetic defect.”).

We think that results of our research cannot be moved to the introduction. We ask to keep the phenotypic description in the results (lines 115 ff.) and the discussion of these results in the discussion (lines 180-190).

(3)

2.5 and 2.5 [2.6 ?] describe the bioinformatics analysis and you cite [16] in both; I think you should provide some details about the pipeline (maybe in a graphical way) so the reader can follow

Response: We added a graphical overview of the bioinformatics pipeline as Figure S1.

(4)

Lines 101-103 can be removed

Response: The reviewer is correct that this sub-chapter 2.7 is redundant with the information in the results where we give the full variant designations according to HGVS nomenclature. However, as reviewer 1 apparently very much liked the explicit mentioning of the reference sequence accession numbers, we suggest to keep this. We leave it up to editorial discretion to remove this small section, if preferred.

(5)

Table 2: Protein-changing variants is not a useful term. Probably "Variants altering the protein sequence"  or something similar.

Response: We apologize if this term is confusing. We need something short to maintain a reasonable layout of the table. We now added a footnote to the table that explains our definition of a “protein-changing” variant in more detail.

(6)

Did you check for any other mutations in the genes associated with other types of GS?

Response: The affected dog did not have any private homozygous variants in either MLPH or RAB27A. This information can be extracted from Table S2. We did not change the manuscript with respect to this comment.